# Eating habit patterns may predict maximum occlusal force: A preliminary study

**Masahiro Okada** [1]*, **Kosuke Okada** [2], **Masayuki Kakehashi** [3]

**1** Department of Food and Dietetics, Hiroshima Bunka Gakuen Two-Year College, Hiroshima, Japan,
**2** Department of Internal Medicine COOP Saeki Hospital, Yahata-higashi, Saeki-ku, Hiroshima, Japan,
**3** Graduate School of Biomedical & Health Sciences, Hiroshima University, Hiroshima, Japan

☯ These authors contributed equally to this work.
* okada@hbg.ac.jp

**Data Availability Statement:** All relevant data are within the manuscript and its Supporting Information files.

**Funding:** The author(s) received no specific funding for this work.

## Abstract

Masticatory function is thought to be related to various eating habits, but it is difficult to evaluate overall masticatory function by assessing complex eating habits. Maximum occlusal force is an important indicator of masticatory function that is affected by age and sex. This preliminary study focused on the maximum occlusal force of young women and their eating habits, excluding food and nutritional factors, and investigated whether individual eating habits and eating habit patterns predict maximum occlusal force. We measured the maximum occlusal force for the whole dentition of 53 healthy young women before they ate lunch. The participants also completed a 12-item questionnaire about their eating habits. Scores were determined from the relationship between each item and increased maximum occlusal force. We found a significant but weak relationship between maximum occlusal force and some questions. The total questionnaire scores for the participants' eating habits showed an almost normal distribution (mean ± standard deviation: 22.7 ± 2.6, median: 23.0, mode: 23.0, range of total scores: 17–28). The accuracy of the linear regression between the total scores for eating habits and maximum occlusal force was high but not perfect ($\beta$: standardized regression coefficient = 0.527, $P < 0.001$). Although further research is needed, our findings confirm that eating habit patterns are significantly associated with maximum occlusal force and may help predict occlusal force. Our results provide important information about eating patterns in humans.

## Introduction

Mastication is essential to daily food intake in humans, and occlusal force is an important indicator in masticatory function [1, 2]. Occlusal force results from the action of the jaw elevator muscles through craniomandibular biomechanics [3]. Studies of occlusal force have approached different relationships at various life stages. In children, occlusal force changes with developing facial morphology and the thickness of the masticatory muscles [4].

The development of occlusal force is related to dietary patterns; however, the relationship with masticatory behavior is unclear [5]. Maximum occlusal force increases with age and is considered stable at approximately 17 years for women and approximately 20 years for men [6]. Maximum occlusal force is generally higher in men than in women because of the larger

**Competing interests:** The authors declare no competing interests.

jaw dimensions and stronger masseter muscles in men [7, 8]. Additionally, body shape factors, such as weight, height, and body mass index (BMI), may influence occlusal force [9].

Maximum occlusal force tends to decrease with age but also varies according to sex. Reduced occlusal force may be an important risk factor in older adults and may be associated with physical activity dysfunction, lower cognitive function, and frailty [10, 11]. It is thought that the maximum occlusal force of older people is related to the status and number of remaining teeth [12]. In the diets of older people, substantial protein, fiber, and vitamin intake may be associated with higher occlusal force [13], and the maintenance of higher maximum occlusal force may improve quality of life (QOL) and prevent various disabilities [13, 14].

There are some studies of the relationship between occlusal force and dietary habits (including food and nutritional factors); however, the relationships between occlusal force at various life stages and detailed eating habits are largely unknown. Some studies have linked mastication and occlusal force with eating habits that do not include food and nutritional factors. In people ranging in age from children to older adults, significant positive correlations were found between masticatory ability and the number of different types of foods consumed during breakfast, but not lunch or dinner [15]. Breakfast habits also affect QOL [16]. In Mexican-American and European-American adults, masticatory ability was significantly correlated with the number of daily meals/snacks [17]. Regular mealtimes may be important for human metabolism. Additionally, a regular mealtime routine may affect mastication [18]. However, the relationship between occlusal force and regular mealtimes is not yet well understood [19]. Masticatory activity at meals is related to the number of chewing cycles and the chewing rate; however, there is no significant correlation between eating behaviors, such as eating speed and efficient chewing, and occlusal force [5, 20, 21]. Additionally, an individual's appetite may change masticatory behavior [22]. In terms of nutritional balance and 'liked and disliked foods', dietary patterns are related to masticatory function and occlusal force; however, eating behavior is not well understood [5, 13, 23]. The relationship between food intake and stress relief is well known [24]. A previous study reported that appropriate chewing produces a greater psychological stress relief effect [25]. Another study suggested that eating with someone and conversing increases occlusal force [26], and conversation is a particularly important factor in increasing the QOL of older people [27].

Eating habits are considered important for masticatory function; however, it may be difficult to evaluate masticatory function by assessing complex eating habits [2]. Detailed investigation of the relationship between masticatory function and eating habits is necessary; therefore, we focused on the relationship between maximum occlusal force and eating habits, excluding food and nutrition factors. To the best of our knowledge, there are few detailed studies of the relationships between maximum occlusal force and eating habits other than those related to dietary content and nutrition. This preliminary study focused on young women, who are thought to more quickly develop a stable maximum occlusal force [28]. We hypothesized that maximum occlusal force in young women could be predicted by analyzing and observing patterns in the relationship between occlusal force and individual eating habits (excluding dietary content and nutrition). We measured the maximum occlusal force of healthy young women immediately before lunch and analyzed the relationships to their individual eating habits as well as to their eating habit patterns. This aim was achieved.

## Materials and methods

### Ethics

This study was approved by the Human Studies Ethical Committee of Hiroshima Bunka Gakuen Two-Year College (approval No.: 22001). This study was part of a previous food intake

and human project (March 2010 to April 2012) [29]. We obtained oral and written informed consent from all participants for inclusion in this study and to undergo the methods used. This study was observational, and all methods were performed in accordance with the principles of the Declaration of Helsinki and the relevant guidelines and regulations.

## Participants

For this study, 53 healthy female Japanese university students participated as volunteers (aged 18–29 years). The inclusion criteria were: non-smokers, not taking prescription medications, and no history of cardiovascular or endocrine disease. The exclusion criteria were excessive weight loss or hospitalization in the previous 3 months. We also excluded patients with pain or discomfort in the mouth, teeth, face, or jaw on the day of measurement. No patients had undergone prosthodontic therapy in the previous 3 months, although some had older prosthodontics. No patients had missing teeth or tooth decay.

We confirmed that the participants were in good health and had fasted (for at least 3 h after breakfast and before lunch) before the measurements were performed. Height and weight were measured with participants wearing indoor clothing with emptied pockets and no shoes. BMI, body fat percentage, and muscle mass were measured using a BC-520 body composition meter (Tanita Corporation, Tokyo, Japan). We used the data set (answers from the questionnaire (S1 Table) and occlusal force values) obtained between 11:30 and 12:00 before lunch. Each participant responded to the questionnaire and had their whole-dentition occlusal force measured while they were alone in a quiet room with appropriate ventilation and lighting. The room temperature was maintained at 20.3 ± 0.9˚C, and all measurements were taken after the participants had adapted to the room temperature for 1 h. Before measuring the occlusal force, we showed the participants the lunch they were to eat later to stimulate their appetite [22]. The lunch we showed all participants was *gyūdon* (beef bowl), consisting of rice and beef. After completing the questionnaire and measurements, all participants ate the *gyūdon*.

## Subjective questionnaire about eating habits

We created an original questionnaire (S1 Table) about eating habits based on studies of QOL, masticatory function, and occlusal force in various lifestyles and life stages [5, 15–17, 19–22, 24–27]. The questionnaire comprised 12 items describing eating habits that were not related to food content (i.e., foodstuff, texture, and nutritional value). Q2 and Q6–10 were answered as "Yes or No". We showed participants the questions and possible responses in the questionnaire, which was administered verbally, and participants selected their responses, which we recorded.

## Measurement of maximum occlusal force

To measure maximum occlusal force, we used a pressure-sensitive film (50H type) and the Dental Prescale System (Fuji Film, Tokyo, Japan). This pressure-sensitive sheet changes color depending on occlusal pressure. We scanned the film and quantified the occlusal force with an Occluzer FPD-707 scanner (GC Corporation, Tokyo, Japan). The occlusal force was measured in Newtons (N), and the value was recorded for each patient's whole dentition. Each participant was seated in a chair with no backrest placed on a flat floor. First, each participant put a test film in their mouth and bit lightly to check the fit of the film. Participants used a medium or large film depending on the fit. Each participant then set the film in their mouth in an appropriate position and bit with maximum occlusal force for 3 s. During measurement, each participant held their head so that the occlusal plane (Frankfurt plane) was parallel to the floor.

The occlusal force was measured twice for each participant, and the higher value was used as the maximum occlusal force.

## Data analysis

SPSS for Windows version 24.0 (IBM SPSS, Tokyo, Japan) was used for all data analysis and figure creation. Descriptive statistics for all participants were expressed as mean ± standard deviation (SD). Multiple regression analysis was used to study the relationship between the individual question responses and the maximum occlusal force. The scores for all items were considered positive for maximum occlusal force and were weighted equally. The total scores for eating habits were calculated using simple addition for each item point, and a histogram was created. Multiple regression analysis was also used to characterize the relationships between maximum occlusal force and the total scores and body composition. Each relationship was analyzed after adjusting for age, height, weight, and BMI [9]. Statistical significance was set at $P < 0.05$.

## Results

Table 1 shows the participants' characteristics, including maximum occlusal force (n = 53). The maximum occlusal force (mean ± SD) was 686.7 ± 300.8 N (range, 98.0–1578.0 N).

Table 2 shows the number of answers to the 12 questions and the mean ± SD maximum occlusal force. Table 3 shows the associations between each of the 12 questions and maximum occlusal force. Among the eating habits, there was a significant association between eating breakfast 'every day' and higher maximum occlusal force ($\beta$: standardized regression coefficient = 0.344, $P = 0.013$). There was also a significant association between the number of meals per day and increasing maximum occlusal force ($\beta = 0.333$, $P = 0.015$). There was a tendency towards higher maximum occlusal force with a 'Yes' response to the questions 'Eat until full' ($\beta = -0.357$, $P = 0.011$) and 'Eat for stress relief' ($\beta = -0.318$, $P = 0.025$). For the other eating habits, there were no significant associations with higher maximum occlusal force. Table 3 also shows the total scores for the eating habits according to the relationships between increased maximum occlusal force and the 12 questions. Higher total eating habit scores were significantly associated with higher maximum occlusal force ($\beta = 0.527$, $P < 0.001$).

The total scores (mean ± standard deviation: 22.7 ± 2.6, median: 23.0, mode: 23.0, range: 17–28) among participants showed an almost normal distribution (Fig 1). Fig 2 is a scatterplot of maximum occlusal force against the total eating habit scores and the simple regression model line (maximum occlusal force = 57.075*×−606.692; $P < 0.001$).

**Table 1. Characteristics of the study population (n = 53).**

| Characteristic | Mean ± SD | Range |
|---|---|---|
| Age (years) | 20.4 ± 2.6 | 18.0–29.0 |
| Height (m) | 1.6 ± 0.1 | 1.5–1.7 |
| Weight (kg) | 52.0 ± 7.4 | 41.7–72.6 |
| Body mass index (kg/m$^2$) | 20.9 ± 3.2 | 16.6–32.9 |
| Body fat percentage (%) | 28.6 ± 5.2 | 17.4–40.9 |
| Muscle mass (kg) | 34.5 ± 3.0 | 28.8–42.2 |
| Maximum occlusal force (N) | 686.7 ± 300.8 | 98.0–1578.0 |

Values are given as mean ± standard deviation.

SD, standard deviation; N, newton.

**Table 2. Questionnaire items and answers.**

| Item | Possible Answers | Value |
|---|---|---|
| 1. Habit of eating breakfast | Skip sometimes, Every day | 19 (546.2 ± 259.8), 34 (765.2 ± 296.8) |
| 2. Always eat at a fixed time | Yes, No | 21 (748.5 ± 326.3), 32 (656.1 ± 280.8) |
| 3. Number of meals per day (including snacks) | 2, 3, 4, 5 | 2 (511.3 ± 67.9), 22 (604.9 ± 281.9), 21 (711.3 ± 320.0), 8 (890.6 ± 248.2) |
| 4. Amount eaten | Small, Medium, Large | 3 (566.6 ± 264.1), 39 (678.5 ± 304.0), 11 (748.2 ± 311.3) |
| 5. Eating speed | Slow, Fast | 22 (641.2 ± 290.5), 31 (718.9 ± 308.6) |
| 6. Chew food well | Yes, No | 22 (695.2 ± 287.0), 31 (680.6 ± 314.8) |
| 7. Eat until full | Yes, No | 20 (814.2 ± 330.2), 33 (609.3 ± 256.9) |
| 8. Think about nutritional balance of the meal | Yes, No | 32 (748.7 ± 314.8), 21 (592.5 ± 257.3) |
| 9. Many likes and dislikes | Yes, No | 13 (661.2 ± 196.0), 40 (694.9 ± 329.5) |
| 10. Eat for stress relief | Yes, No | 34 (765.1 ± 306.2), 19 (546.3 ± 239.2) |
| 11. Eat with others or alone (including family) | Alone, Sometimes eat with others, Always eat with others | 2 (590.3 ± 58.2), 14 (589.2 ± 292.9), 37 (728.8 ± 306.0) |
| 12. Conversation when eating | No conversation, Sometimes conversation, Always conversation | 7 (602.7 ± 133.2), 33 (659.2 ± 327.7), 13 (801.6 ± 276.4) |

The value indicates the number of answers (mean ± standard deviation maximum occlusal force).

In the S2 Table, we showed the relationships between maximum occlusal force and body composition. Higher participants' maximum occlusal force was associated with higher muscle mass ($\beta = 0.367$, $P = 0.035$).

## Discussion

We investigated maximum occlusal force and eating habits in healthy young women. Our findings revealed a weak but significant relationship between maximum occlusal force in healthy young women and breakfast habits, number of meals per day, 'eat until full', and 'eat

**Table 3. Relationship between maximum occlusal force and eating habits and the resulting eating habit scores.**

| Item | β (P) | Scores |
|---|---|---|
| 1. Habit of eating breakfast | 0.344 (0.013) | 1 or 2 |
| 2. Always eat at a fixed time | −0.120 (0.408) | 1 or 2 |
| 3. Number of meals per day (including snacks) | 0.333 (0.015) | 2–5 |
| 4. Amount eaten | 0.141 (0.317) | 1–3 |
| 5. Eating speed | 0.110 (0.452) | 1 or 2 |
| 6. Chew food well | 0.020 (0.891) | 1 or 2 |
| 7. Eat until full | −0.357 (0.011) | 1 or 2 |
| 8. Think about nutritional balance of the meal | −0.228 (0.109) | 1 or 2 |
| 9. Many likes and dislikes | −0.003 (0.981) | 1 or 2 |
| 10. Eat for stress relief | −0.318 (0.025) | 1 or 2 |
| 11. Eat with others or alone (including family) | 0.173 (0.245) | 1–3 |
| 12. Conversation when eating | 0.213 (0.171) | 1–3 |
| Total eating habit scores | 0.527 (<0.001) | 17–28 |

Analysis of individual items was performed after adjusting for age, height, and weight.

The value in parentheses indicates the *P* value.

β: standardized regression coefficient.

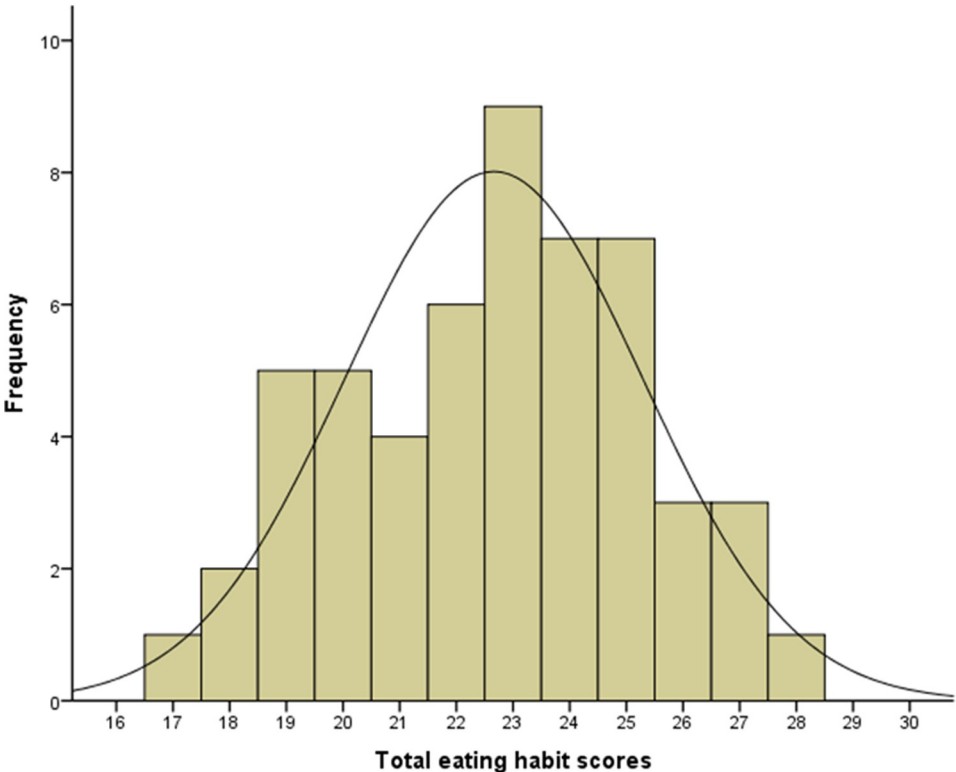

**Fig 1. Total scores (mean ± standard deviation: 22.7 ± 2.6, median: 23.0, mode: 23.0, range: 17–28) among participants, showing an almost normal distribution.**

for stress relief'. Higher maximum occlusal force was strongly associated with eating habit patterns, suggesting that occlusal force is not determined by a particular eating habit, but by overall eating habit patterns.

In this study, we created an original simple eating habit questionnaire that excluded the factors of dietary content and the nutritional status of the food in our investigation of maximum occlusal force.

The average age of the 53 women in our study was (mean ± SD) 20.4 ± 2.6 years (range: 18.0–29.0 years), indicating that all participants were at the life stage with the highest and most stable maximum occlusal force [28].

There was a significant relationship between the habit of eating breakfast and maximum occlusal force. We also found a significant relationship between the number of meals per day and maximum occlusal force. Most participants in our research ate three meals (breakfast, lunch, dinner) or four meals including snacks per day; maximum occlusal force tended to increase as the number of meals increased. Tooth wear, which is closely related to occlusal force, is significantly correlated with the number of daily meals/snacks [17]. We suggest that the development of occlusal force is more closely related to occlusal habits, such as eating breakfast and the number of daily meals than to eating at a fixed time.

We expected to find a relationship between 'Eating speed' and 'Chew food well' and occlusal force; however, no significant relationships were found. Previous studies of Japanese children also found no correlation between 'Chewing speed' and 'Chew food well' and occlusal force [5, 20]. We also found no relationship between 'Amount eaten' and occlusal force. As in previous studies, our results suggest that these habits relating to masticatory behavior may not be strongly associated with the development of occlusal force.

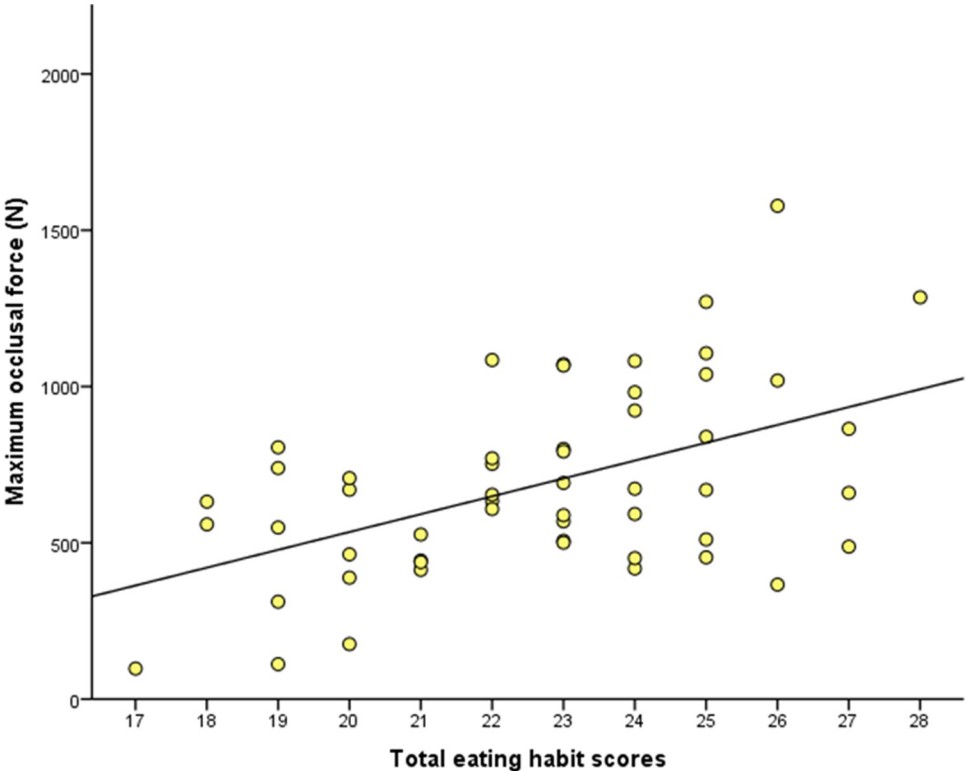

**Fig 2. Scatterplot showing the relationship between maximum occlusal force and total eating habit scores.** The regression line was calculated by simple regression analysis (maximum occlusal force = 57.075*×−606.692; $P < 0.001$).

There was a significant relationship between 'Eat until full' and maximum occlusal force. Consciousness of eating until you are full may result from having a strong appetite, and the human appetite may be closely related to masticatory function [22]. We consider that the development of human appetite is an important factor in the development of occlusal force.

We found no relationship between 'Think about the nutritional balance of the meal' or 'Many likes and dislikes' and maximum occlusal force. Actual dietary patterns may be more important than consciousness of factors such as nutritional balance and food preference in the development of occlusal force [5, 23].

There was a significant relationship between 'Eat for stress relief' and maximum occlusal force. In older people, eating and conversation are particularly important factors in improving QOL and oral function [27]. There were no significant relationships between 'Eating with others or alone' and 'Conversation when eating' and maximum occlusal force; however, we suggest that these items may be involved in stress reduction and the development of occlusal force.

There was a tendency for each eating habit to have a correlation with maximum occlusal force, and some significant relationships were found. In previous studies, the relationship between objective occlusal function and eating habits could not be assessed by existing subjective dietary questionnaires [2, 20]. Using our questionnaire, we found a strong relationship between the total eating habit pattern scores and maximum occlusal force. Although the relationships between each eating habit and maximum occlusal force were not strong, we suggest that participants' overall eating habit patterns are strongly related to the development of occlusal force. Occlusal force is not maintained by transient masticatory training [30]; therefore, we

suggest that the participants' maximum occlusal force was maintained because of long-term eating habits.

Masticatory function and occlusal force have also drawn attention in research of obesity and body composition [31, 32]. Our data for the relationship between maximum occlusal force and muscle mass are shown in S2 Table. It is thought that there is a relationship between body growth and the development of occlusal force [31]. In younger people, height, weight, and BMI may be related to maximum occlusal force [9]. We suggest that there is a relationship between muscle growth and the development of occlusal force; however, more research is needed to understand the relationship between maximum occlusal force and body composition and eating habits.

Although the results of our research are important, there were limitations. First, the sample size in this preliminary study was small, with only 53 women participating. Because the development of occlusal force is influenced by sex [6–8, 28], an analysis of men is needed to determine the relationship between occlusal force and eating habits in this sex. The effect of occlusal force differs by life stage. Many researchers pay particular attention to the development of occlusal force in children and the maintenance of occlusal force in older people [4, 5, 10–14, 20, 23, 26, 28, 30]. We suggest that further research should survey occlusal force and eating habits in a larger sample at various life stages. Second, the measurement of maximum occlusal force may require consideration of bio-directional confounders, such as craniofacial morphology and sleep bruxism, as well as malocclusion, muscle mass/strength, TMJ dysfunction, and grinding/clenching, which were not examined in our study [31, 33]. Third, the questionnaire we created may require further factor analysis between questions. Our questions were not subject to sufficient psychological validation; therefore, it may be necessary to review and improve the questionnaire for use in further research. Fourth, eating habits may differ between Japan and other countries. Furthermore, maximum occlusal force was the only factor we investigated as an indicator of masticatory function. Further research into other factors is required to better understand masticatory function in humans. Despite these limitations, our findings provide useful information about human occlusion and eating habits.

In conclusion, although further research is needed, our findings confirm that eating habit patterns are significantly associated with maximum occlusal force and may help predict occlusal force. Our results provide important information about eating patterns in humans.

## Supporting information

**S1 Table. Eating habit questionnaire.**
(DOCX)

**S2 Table. Relationships between maximum occlusal force and body composition.**
(DOCX)

**S1 Dataset. Minimal data set.**
(CSV)

## Acknowledgments

We thank Hiroshima Bunka Gakuen University for lending the analytical equipment. We thank the staff of Hiroshima University School of Dentistry for providing appropriate advice about our research. We also thank Helen Jeays, BDSc AE, and Jane Charbonneau, DVM, from Edanz (https://jp.edanz.com/ac) for editing a draft of this manuscript.

## Author Contributions

**Formal analysis:** Masahiro Okada.

**Investigation:** Masahiro Okada.

**Project administration:** Masahiro Okada.

**Writing – original draft:** Masahiro Okada.

**Writing – review & editing:** Kosuke Okada, Masayuki Kakehashi.

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
