## [Decision Letter · Decision Letter 0]

25 Nov 2021

PONE-D-20-39854Eating habit patterns may help predict maximum occlusal force in healthy young women: a preliminary studyPLOS ONE

Dear Dr. Okada,

Thank you for submitting your manuscript to PLOS ONE. After careful consideration, we feel that it has merit but does not fully meet PLOS ONE’s publication criteria as it currently stands. Therefore, we invite you to submit a revised version of the manuscript that addresses the points raised during the review process.

 Three reviewers have evaluated your submission, and have identified several aspects of the study design that require further clarification to ensure reproducibility. They have also pointed to aspects of the presentation of the manuscript that require improvement in order to meet PLOS ONE's publication criteria. Please respond carefully to all of the reviewers' concerns when preparing your revisions.

We look forward to receiving your revised manuscript.

Kind regards,

Jamie Males

Staff Editor

PLOS ONE

Journal Requirements:

"This study was partly supported by Hiroshima Bunka Gakuen University. We thank the staff of Hiroshima University School of Dentistry for providing appropriate advice about our research. 

We also thank Helen Jeays, BDSc AE, from Edanz Group (https://en-author-services.edanz.com/ac) for editing a draft of this manuscript."

"This study was partly supported by Hiroshima Bunka Gakuen University."

Reviewers' comments:

Reviewer's Responses to Questions

**Comments to the Author**

1. Is the manuscript technically sound, and do the data support the conclusions?

Reviewer #1: Yes

Reviewer #2: No

Reviewer #3: Partly

2. Has the statistical analysis been performed appropriately and rigorously? 

Reviewer #1: Yes

Reviewer #2: Yes

Reviewer #3: Yes

3. Have the authors made all data underlying the findings in their manuscript fully available?

Reviewer #1: Yes

Reviewer #2: Yes

Reviewer #3: Yes

4. Is the manuscript presented in an intelligible fashion and written in standard English?

Reviewer #1: Yes

Reviewer #2: Yes

Reviewer #3: Yes

5. Review Comments to the Author

Reviewer #1: The authors in this study titled “Eating habit patterns may help predict maximum occlusal force in healthy young women: a preliminary study” aimed to analyze the eating habit patterns and occlusal force patterns in young women. This is an interesting study; however, I have few concerns that I have highlighted below:

Introduction: The importance of occlusal forces and its relevance in clinical medicine and dentistry should be described, particularly, the influence of occlusal forces on prosthodontic rehabilitation of teeth.

Methods: Kindly indicate if any of the patients had ever received any restorations. This is not mentioned in the inclusion criteria.

Results: Appropriate. Figure 2 that shows the linear relationship between maximum occlusal force and eating habits was useful.

Discussion: As mentioned earlier, the authors should focus on the importance occlusal forces on prosthodontic rehabilitation and restorations. This will be helpful for the clinicians whilst planning restorations or replacement of missing teeth.

Reviewer #2: This study is the first to investigate the association between maximum occlusal force and eating habits in young women. Although the authors do present some new findings, a number of clarifications need to be made before the conclusion can be drawn.

1. How the maximum occlusal force is determined? Did the authors measure the whole dentition or molar area? The bite forces vary significantly in the oral cavity.

2. Also, there are various confounding factors, such as malocclusion, muscle mass/strength, TMJ dysfunction, grinding/clenching habits etc. All these factors may affect participants’ eating habits and/or maximum occlusal forces.

Reviewer #3: Manuscript title: Eating habit patterns may help predict maximum occlusal force in

healthy young women: a preliminary study.

This is an interesting topic and I would like to thank the authors for the great effort and time they spend on this paper. However, I have some concerns and hope the authors will address them before recommending this manuscript for publication.

Title: The title is somewhat narrative. Please rewrite it to make it more vital, concise, attract the reader, and come to the point.

Introduction:

1. The introduction is too long.

2. You should start with a brief introduction of the topic, then write the body of the introduction by focusing on the previous studies related to your topic. After that tell the readers why you are conducting this study and why your research is important, and finally, conclude with your aims and objectives.

Methos:

1. Too long and repetitive

2. Please provide clear inclusion and exclusion criteria at the beginning of this section.

3. Subjective questionnaire about eating habits: Give a brief description of the questionnaire. You needn’t write every single question in this section. You can add the questionnaire at the end of this paper as an index or supplementary file.

Discussion:

1. The discussion is also too long.

2. In lines from 251 to 255, you wrote ‘’ there was a significant but not strong relationship between some eating habits…… maximum occlusal force in healthy young women’’. Which eating habits were significant? Don’t force the reader to search about these significant habits across the manuscript. Please add it.

3. Lines 259 to 277 are repetitive to the introduction section. Please delete it.

4. In lines 278 to 280, you wrote ‘’ Additionally, we created an original simple eating habit questionnaire that excluded the factors of dietary ……. investigations of maximum occlusal force.’’ My question is, have you tried to validate your questionnaire before commencing this study or not?

Conclusion:

• Please provide an accurate and concise conclusion that coincides with that in the abstract.

References:

There are 47 references. I advise the authors to reduce it as much as they can by selecting the most relevant ones.

6. PLOS authors have the option to publish the peer review history of their article (what does this mean?). If published, this will include your full peer review and any attached files.

Reviewer #1: **Yes: **Jayakumar Jayaraman

Reviewer #2: No

Reviewer #3: No

---

## [Author Response · Author response to Decision Letter 0]

28 Dec 2021

Jamie Males

Staff Editor

PLOS ONE

28 December 2021

Dear Dr Males,

Re: Resubmission of manuscript reference No. PONE-D-20-39854

Please find attached a revised version of our manuscript, originally titled “Eating habit patterns may help predict maximum occlusal force in healthy young women: a preliminary study”, (now titled “Eating habit patterns may predict maximum occlusal force”) which we would like to submit for consideration for publication as a Research Article in PLOS ONE. 

Your comments and those of the reviewers were highly insightful and enabled us to greatly improve the quality of our manuscript. In the following pages are our point-by-point responses to each of the comments of the reviewers as well as your own comments.

As requested, we are submitting a copy of our revised manuscript with the tracked changes visible and another as a clear copy without the tracked changes. Additions and changes to the text in the manuscript appear in red font. Deleted text is visible in the copy with the tracked changes.

We hope that the revisions in the manuscript and our accompanying responses will be sufficient to make our manuscript suitable for publication in PLOS ONE.

We look forward to hearing from you at your earliest convenience.

Yours sincerely,

Masahiro Okada

Department of Food and Dietetics

Hiroshima Bunka Gakuen Two-Year College

Hiroshima, Japan

Tel.: +81-50-3535-1351

Fax: +81-82-239-2863

E-mail: okada@hbg.ac.jp

 

Responses to the Staff Editor:

Comment 1: “Please ensure that your manuscript meets PLOS ONE's style requirements, including those for file naming. The PLOS ONE style templates can be found at 

https://journals.plos.org/plosone/s/file?id=ba62/PLOSOne_formatting_sample_title_authors_affiliations.pdf”

Response: In response to the editor’s comment, our manuscript has been formatted in accordance with the journal’s title page and manuscript templates.

Comment 2: “Thank you for stating the following in the Acknowledgments Section of your manuscript: 

"This study was partly supported by Hiroshima Bunka Gakuen University. We thank the staff of Hiroshima University School of Dentistry for providing appropriate advice about our research. 

We also thank Helen Jeays, BDSc AE, from Edanz Group (https://en-author-services.edanz.com/ac) for editing a draft of this manuscript."

"This study was partly supported by Hiroshima Bunka Gakuen University."

Please include your amended statements within your cover letter; we will change the online submission form on your behalf.”

Response: We would like to change our funding statement to the following: “The authors received no specific funding for this work.” Accordingly, we have revised the previous statement that you noted in the Acknowledgments from “This study was partly supported by Hiroshima Bunka Gakuen University.” to “We thank Hiroshima Bunka Gakuen University for lending the analytical equipment.”

Comment 3: “In your Data Availability statement, you have not specified where the minimal data set underlying the results described in your manuscript can be found. PLOS defines a study's minimal data set as the underlying data used to reach the conclusions drawn in the manuscript and any additional data required to replicate the reported study findings in their entirety. All PLOS journals require that the minimal data set be made fully available. For more information about our data policy, please see http://journals.plos.org/plosone/s/data-availability.

We will update your Data Availability statement to reflect the information you provide in your cover letter.”

Response: We are submitting our minimal data set as a csv file: “S3 minimal data set”, as supporting information. There are no ethical or legal restrictions to sharing our data publicly.

 

Responses to the comments of Reviewer #1:

Comment 1:”The authors in this study titled “Eating habit patterns may help predict maximum occlusal force in healthy young women: a preliminary study” aimed to analyze the eating habit patterns and occlusal force patterns in young women. This is an interesting study; however, I have few concerns that I have highlighted below:”

Response: We thank Reviewer 1 for these comments. Our responses to the reviewer’s additional comments follow.

Comment 2: “Introduction: The importance of occlusal forces and its relevance in clinical medicine and dentistry should be described, particularly, the influence of occlusal forces on prosthodontic rehabilitation of teeth.”

Response: Occlusal force is a necessary factor for masticatory function, and weak occlusal force may affect masticatory function as well as food intake. Long-term, occlusal force may be related to improving physical and cognitive function (nervous system), and quality of life. Of course, occlusal force is greatly related to a person’s oral health and is an important index in human health from the viewpoint of both medicine and dentistry. Changes in occlusal force across life stages are recognized. Increasing maximum occlusal force at its peak in early adulthood may contribute to maintaining health throughout adulthood. We discussed these points in the following text in the introduction:

Page 2, lines 41–42: “Mastication is essential to daily food intake in humans, and occlusal force is an important indicator in masticatory function [1, 2].”

Page 3, lines 55−57: “Reduced occlusal force may be an important risk factor in older adults and may be associated with physical activity dysfunction, lower cognitive function, and frailty [10, 11].”

Page 3, lines 58−61: “In the diets of older people, substantial protein, fiber, and vitamin intake may be associated with higher occlusal force [13], and the maintenance of higher maximum occlusal force may improve quality of life (QOL) and prevent various disabilities [13, 14].”

Page 4, lines 71−74: “Regular mealtimes may be important for human metabolism. Additionally, a regular mealtime routine may affect mastication [18]. However, the relationship between occlusal force and regular meal times is not yet well understood [19].”

Regarding prosthodontics, this was not a focus of our study. All of the participants were women in their 20s. None of the participants underwent dental prosthodontic treatment during the previous 3 months, and none had missing teeth. Some participants had old prosthodontics, but participants with pain or discomfort in the mouth, teeth, face, or jaw on the day of measurement were excluded from this study. However, we agree that the relationship between occlusal force and prosthodontics may be important, and this is an area of future research.

We added text discussing the lack of prosthodontic therapy on Page 6, lines 116–118:

“No patients had undergone prosthodontic therapy in the previous 3 months, although some had older prosthodontics. No patients had missing teeth or tooth decay.”

Comment 2: “Methods: Kindly indicate if any of the patients had ever received any restorations. This is not mentioned in the inclusion criteria.”

Response: None of the participants underwent dental prosthodontic treatment during the previous 3 months, and none had missing teeth. Some participants had old prosthodontics, but participants with pain or discomfort in the mouth, teeth, face, or jaw on the day of measurement were excluded from this study.

Comment 3: “Results: Appropriate. Figure 2 that shows the linear relationship between maximum occlusal force and eating habits was useful.”

Response: We thank Reviewer #2 for these comments.

Comment 4: ”Discussion: As mentioned earlier, the authors should focus on the importance occlusal forces on prosthodontic rehabilitation and restorations. This will be helpful for the clinicians whilst planning restorations or replacement of missing teeth.”

Response: We agree with the reviewer that occlusal forces are important in prosthodontic rehabilitation and restorations. However, this was not the focus of our study and is a potential area of future research. 

Responses to the comments of Reviewer #2:

Initial Comment: “This study is the first to investigate the association between maximum occlusal force and eating habits in young women. Although the authors do present some new findings, a number of clarifications need to be made before the conclusion can be drawn.”

Response: We thank Reviewer #2 for these comments. Our responses to the reviewer’s additional comments follow.

Comment 1: “How the maximum occlusal force is determined? Did the authors measure the whole dentition or molar area? The bite forces vary significantly in the oral cavity.”

Response: We described the method of measuring maximum occlusal force in the revised manuscript, in the section titled “Measurement of maximum occlusal force.” However, to clarify, 

we measured the occlusal force for each participant’s whole dentition. We added the following:

Page 7, lines 149–150: “The occlusal force was measured in Newtons (N), and the value was recorded for each patient’s whole dentition.”

Comment 2: “Also, there are various confounding factors, such as malocclusion, muscle mass/strength, TMJ dysfunction, grinding/clenching habits etc. All these factors may affect participants’ eating habits and/or maximum occlusal forces.”

Response: We thank the reviewer for this comment. We agree that confounding factors may have affected the participants’ eating habits and/or maximum occlusal forces. We discussed that we did not examine confounding factors as a limitation in the original discussion. However, we added to this statement in the revised manuscript, as follows (Page 19, lines 289–292):

“Second, the measurement of maximum occlusal force may require consideration of bio-directional confounders, such as craniofacial morphology and sleep bruxism, as well as malocclusion, muscle mass/strength, TMJ dysfunction, and grinding/clenching, which were not examined in our study [31].”

 

Responses to the comments of Reviewer #3:

Comment 1: “Manuscript title: Eating habit patterns may help predict maximum occlusal force in

healthy young women: a preliminary study.

This is an interesting topic and I would like to thank the authors for the great effort and time they spend on this paper. However, I have some concerns and hope the authors will address them before recommending this manuscript for publication.

Title: The title is somewhat narrative. Please rewrite it to make it more vital, concise, attract the reader, and come to the point.”

Response: We thank Reviewer #3 for these comments. To address the reviewer’s concern regarding the title, we changed the title to the following: “Eating habit patterns may predict maximum occlusal force” (title page).

Comment 2: 

“Introduction:

1. The introduction is too long.

2. You should start with a brief introduction of the topic, then write the body of the introduction by focusing on the previous studies related to your topic. After that tell the readers why you are conducting this study and why your research is important, and finally, conclude with your aims and objectives.”

Response: To address the reviewer’s concerns, we shortened the introduction by deleting several sections of text and focusing on previous studies, why we conducted the study, why our research is important, and ending with our aims and objectives.

Comment 3: Comments regarding the Methods:

“1. Too long and repetitive”

Response: We have revised the Methods in accordance with the reviewer’s recommendation by deleting repetitive text and revising for conciseness.

“2. Please provide clear inclusion and exclusion criteria at the beginning of this section.”

Response: We added the following section describing the inclusion and exclusion criteria (Page 5-6, lines 112–118):

“The inclusion criteria were: non-smokers, not taking prescription medications, and no history of cardiovascular or endocrine disease. The exclusion criteria were excessive weight loss or hospitalization in the previous 3 months. We also excluded patients with pain or discomfort in the mouth, teeth, face, or jaw on the day of measurement. No patients had undergone prosthodontic therapy in the previous 3 months, although some had older prosthodontics. No patients had missing teeth or tooth decay.”

“3. Subjective questionnaire about eating habits: Give a brief description of the questionnaire. You needn’t write every single question in this section. You can add the questionnaire at the end of this paper as an index or supplementary file.”

Response: To address the reviewer’s concern, we deleted the list of the questions from the methods. This information is now included as the full questionnaire as Supplemental Table 1 (S1 Table).

Comment 4: Comments regarding the discussion:

“1. The discussion is also too long.”

Response: To address the reviewer’s concern, we deleted several sections of text in the discussion to avoid repetition, and we revised for conciseness.

“2. In lines from 251 to 255, you wrote ‘’ there was a significant but not strong relationship between some eating habits…… maximum occlusal force in healthy young women’’. Which eating habits were significant? Don’t force the reader to search about these significant habits across the manuscript. Please add it.”

Response: We thank the reviewer for this question and comment. To clarify, we found significant but weak relationships with breakfast habits, number of meals, 'eat until full', ‘eat for stress relief’ and occlusal force. These factors, including breakfast habits, may be important; however, they may be only minor factors that affect human occlusal force. We revised the relevant section of the discussion to clarify and to emphasize the suggestion that occlusal force is not determined by a particular eating habit, but by overall eating habit patterns: (Page 16, lines 219–223)

“Our findings revealed a weak but significant relationship between maximum occlusal force in healthy young women and breakfast habits, number of meals per day, ‘eat until full’, and ‘eat for stress relief’. Higher maximum occlusal force was strongly associated with eating habit patterns, suggesting that occlusal force is not determined by a particular eating habit, but by overall eating habit patterns.”

“3. Lines 259 to 277 are repetitive to the introduction section. Please delete it.”

Response: We have deleted lines 259 to 277.

“4. In lines 278 to 280, you wrote ‘’ Additionally, we created an original simple eating habit questionnaire that excluded the factors of dietary ……. investigations of maximum occlusal force.’’ My question is, have you tried to validate your questionnaire before commencing this study or not?”

Response: We thank the reviewer for this comment. Our questionnaire items were not subject to sufficient psychological validation prior to this study. We included this point as a limitation in the original discussion (Page 19, lines 292–295 (revised manuscript)):

“Third, the questionnaire we created may require further factor analysis between questions. Our questions were not subject to sufficient psychological validation; therefore, it may be necessary to review and improve the questionnaire for use in further research.”

Comment 5:

“Conclusion:

• Please provide an accurate and concise conclusion that coincides with that in the abstract.”

Response: To address the reviewer’s comment, we deleted the final paragraph of the original discussion and added the following conclusions to match those in the abstract: (Page 19, lines 301–304)

“In conclusion, although further research is needed, our findings confirm that eating habit patterns are significantly associated with maximum occlusal force and may help predict occlusal force. Our results provide important information about eating patterns in humans.”

“References:

There are 47 references. I advise the authors to reduce it as much as they can by selecting the most relevant ones.”

Response: To address the reviewer’s comment, we reduced the number of references from 47 to 33 (we added one new reference, #18), selecting the most relevant references.

---

## [Decision Letter · Decision Letter 1]

25 Jan 2022

Eating habit patterns may predict maximum occlusal force

PONE-D-20-39854R1

Dear Dr. Okada,

We’re pleased to inform you that your manuscript has been judged scientifically suitable for publication and will be formally accepted for publication once it meets all outstanding technical requirements.

Kind regards,

Carla Pegoraro

Division Editor

PLOS ONE

Additional Editor Comments (optional):

Please amend the title of the study to include 'preliminary study' as suggested by two of the reviewers during the last technical checks prior to final publication.

Reviewers' comments:

Reviewer's Responses to Questions

**Comments to the Author**

1. If the authors have adequately addressed your comments raised in a previous round of review and you feel that this manuscript is now acceptable for publication, you may indicate that here to bypass the “Comments to the Author” section, enter your conflict of interest statement in the “Confidential to Editor” section, and submit your "Accept" recommendation.

Reviewer #1: All comments have been addressed

Reviewer #2: All comments have been addressed

Reviewer #3: (No Response)

2. Is the manuscript technically sound, and do the data support the conclusions?

Reviewer #1: Yes

Reviewer #2: Yes

Reviewer #3: Yes

3. Has the statistical analysis been performed appropriately and rigorously? 

Reviewer #1: Yes

Reviewer #2: Yes

Reviewer #3: Yes

4. Have the authors made all data underlying the findings in their manuscript fully available?

Reviewer #1: Yes

Reviewer #2: Yes

Reviewer #3: Yes

5. Is the manuscript presented in an intelligible fashion and written in standard English?

Reviewer #1: Yes

Reviewer #2: Yes

Reviewer #3: Yes

6. Review Comments to the Author

Reviewer #1: The authors have adequately addressed my comments raised in the previous review. My only suggestion is to retain "preliminary study" in the title considering the small sample size focussing on women.

Reviewer #2: (No Response)

Reviewer #3: I thank the authors for their significant efforts in addressing all the reviewers' concerns. However, I am not satisfied with the title and ask the authors to think about it again. I suggest the following title '' Relationship between eating habit patterns and maximum occlusal force: a preliminary study.

7. PLOS authors have the option to publish the peer review history of their article (what does this mean?). If published, this will include your full peer review and any attached files.

Reviewer #1: **Yes: **Jayakumar Jayaraman

Reviewer #2: No

Reviewer #3: No

---

## [Editor Report · Acceptance letter]

7 Feb 2022

PONE-D-20-39854R1 

Eating habit patterns may predict maximum occlusal force: a preliminary study 

Dear Dr. Okada:

I'm pleased to inform you that your manuscript has been deemed suitable for publication in PLOS ONE. Congratulations! Your manuscript is now with our production department. 

Kind regards, 

on behalf of

Dr Carla Pegoraro 

Staff Editor

PLOS ONE